materials science/nanotechnology

air filter, superfine glass fibre, photocatalyst, antibacterial, visible light

**Author for correspondence:**
Fuqiang Zhai
e-mail: zhaifuqiangupc@163.com

# A superfine glass fibre air filter with rapid response to photocatalytic antibacterial properties under visible light by loading rGO/ZnO

Yongyi Luo[1], Fuqiang Zhai[3], Yingchun Zhang[2], Zhiqian Chen[1], Mingde Ding[3], Dajiang Qin[5], Jinming Yang[5], Guang Feng[4] and Lu Li[3]

[1]School of Materials and Energy, and [2]College of Pharmaceutical Sciences, Southwest University, Chongqing 402160, People's Republic of China
[3]Micro/Nano Optoelectronic Materials and Devices International Science and Technology Cooperation Base of China, Chongqing University of Arts and Sciences, Chongqing 402160, People's Republic of China
[4]Engineering Research Center of Optical Instrument and System, Chongqing Institute of East China Normal University, Chongqing 401120, People's Republic of China
[5]Chongqing Zisun Technology Co., Ltd., Chongqing 401120, People's Republic of China

 FZ, 0000-0002-7634-2723

The development of high-performance air filter has become more and more important to public health. However, it has always been very challenging for developing a multifunctional air filter to simultaneously achieve excellent filtration and antibacterial properties. Herein, a versatile air filter was prepared with loading the reduced graphene (rGO) and zinc oxide on the superfine glass fibre (s-GF) with the three-dimensional network structure by *in situ* sol–gel process followed by calcination, which aims to achieve synergistic high-efficiency air filtration and rapid response to photocatalytic antibacterial properties under visible light. The air filter showed a three-dimensional network structure based on a rGO/ZnO/s-GF multilayer and exhibited the highest catalytic performance by achieving a 95% degradation effect on rhodamine B within 2 h and achieving 100% antibacterial inactivation of the *Escherichia coli* and *Staphylococcus aureus* within 4 h under visible light when the weight ratio of rGO in rGO/ZnO is 1.6%. The air filtration efficiency can also be maintained at 99% after loading ZnO and rGO photocatalytic particles. The spectrum of the photoluminescence (PL), UV-Vis diffuse reflectance spectra (DRS) and electron spin resonance (ESR) indicate that the combination of rGO and

ZnO on the s-GF can increase the separation of photogenerated carriers and the specific surface area of the air filter, thereby increasing the photocatalytic response and antibacterial properties of the s-GF air filter under visible light in a short time.

## 1. Introduction

Semiconductor photocatalysis has been widely regarded as one of the most promising technologies in air purification especially in high population density occasions where the antibacterial air filtration is urgently required [1,2]. What is more, the superfine glass fibre (s-GF) has better physical properties in the field of air filtration and thermal-acoustic insulation due to its small diameter lower than 4 µm and strong interweaving network than the traditional fibre materials, resulting in that it has been widely used as filter elements for air purification [3]. The main filtering principle of s-GF filter is to capture the fine particulate pollutants through interception and electrostatic adsorption through its three-dimensional network structure. Although the high-efficiency air purifier can capture most of bacteria and viruses, it cannot kill the captured microorganisms. Studies show that some microorganisms can survive on the filter for a long time in a continuously ventilated and dry pipe environment. When the ventilation is closed as well as the suitable air humidity–temperature and wind speed, the attached microorganisms on the filter surface can also multiply and grow, which can cause some human diseases, such as allergies, infectious, biological toxicity, etc. [4,5]. It has always been a challenge for developing an efficient multifunctional air filter to simultaneously achieve air filtration and antibacterial properties. Therefore, the development of a novel filter to combine the s-GF and photocatalytic semiconductor materials is a promising way for air purification and sterilization in the densely populated environment.

Photocatalysis as one of the effective means to degrade harmful gases and antibacterial in the air has been confirmed by a mount of researches [6–8]. Zinc oxide (ZnO) is widely used in photocatalysis due to its environmental friendliness and low cost [9]. But ZnO also shows the following two shortcomings: (i) its wide band gap and strict requirement for light source corresponding to the ultraviolet light and wavelength below 380 nm results in it being impossible to be used under visible light [10]; (ii) its high photogenerated electron–hole pair recombination rate, which leads to the ZnO modification becoming a hot spot in current photocatalytic performance research in order to achieve rapid visible light response and photocatalytic antibacterial effect of ZnO [11–13].

Currently, studies have shown that the addition of reduced graphene (rGO) can improve the performance of ZnO by enhancing the photocatalytic response of ZnO under visible light [14,15]. The larger specific surface area of the rGO/ZnO composite can provide more active sites so that it can capture more microorganisms. In addition, rGO can also promote charge transfer and the separation of photogenerated electron–hole pairs [16]. Furthermore, adding ZnO nanoparticles can also inhibit the agglomeration between rGO layered structures to a certain extent [17]. Meanwhile, many studies have also proved that loading photocatalytic particles on a carrier [18,19] or preparing photocatalysts into nanofibres by electrospinning [20,21] may be a worthwhile method to try due to its high-efficiency photocatalytic characteristics and environmental friendliness. In comparison, there is little research on making the best of the three-dimensional network structure of s-GF for simultaneous air filtration and antibacterial testing in actual application. As a result, loading rGO and ZnO on s-GF air filter may be an effective method to improve the photocatalytic antibacterial performance under visible light conditions.

In this study, a multifunctional s-GF air filter with rapid response to photocatalytic antibacterial properties under visible light by loading rGO/ZnO has been successfully synthesized. The morphology, structure, specific surface area and composition of the obtained rGO/ZnO/s-GF air filter were characterized. In addition, the photocatalytic degradation mechanism of rhodamine B (RhB) solution and the antibacterial effect of *Escherichia coli* and *Staphylococcus aureus* in a short time were studied under the visible light condition. Besides, the mechanism of the enhanced photocatalytic antibacterial properties of the multifunctional air filter was also discussed.

## 2. Material and methods

### 2.1. Materials

The rGO solution with a 3% solid content was purchased from Daying Juneng Technology and Development Co., Ltd. Zinc acetate dihydrate (Zn (CH$_3$COO)$_2$ · 2H$_2$O) was bought from Macklin chemicals and the

**Figure 1.** Schematic diagram of the formation of rGO/ZnO/s-GF air filter.

ammonia solution ($NH_3 \cdot H_2O$) was from Chengdu Kelong Chemical Co., Ltd. The superfine glass fibre (s-GF) air filter paper with a H13 filtration grade was provided from Zisun Technology Co., Ltd. All the above-mentioned chemicals were of analytical grade and used without further purification.

## 2.2. Synthesis of the rGO/ZnO/s-GF air filter

The multifunctional rGO/ZnO/s-GF air filter was synthesized by *in situ* sol–gel process followed by calcination [22]. The specific diagram of rGO/ZnO/s-GF air filter synthesis is shown in figure 1. Firstly, the s-GF paper with a H13 filtration grade was prepared by combining different diameters s-GF to weave a strong and stable fibre three-dimensional network structure and then it was cut into a size of $4 \times 3.5$ cm for next step use. Secondly, the process of synthesis precursor solution of zinc oxide was as follows: 4 g of zinc acetate dihydrate was dissolved in 50 ml of distilled water and then 3 ml of ammonia was added to the solution to form the zinc–ammonia complex precipitate. The formed precipitate was extracted by the suction filtration and rinsed with a large amount of distilled water and alcohol and then it was stored at 60°C for drying 10 h. In order to obtain the 0.8 and 1.6 mol l$^{-1}$ ZnO precursor solutions, 3.4 and 6.8 g of the prepared precipitate were weighed and mixed with 20 ml distilled water and 20 ml ammonia water, respectively. Then 1, 2 and 3 ml rGO solution were diluted in 50 ml distilled water and the three kinds of diluted solutions were ultrasonicated for 0.5 h at room temperature to obtain a uniformly dispersed rGO solution, respectively. Furthermore, the above-prepared s-GF air filter paper was immersed in the dispersion and kept in it for 5 min to make rGO evenly dispersed on the surface of the s-GF air filter paper by ultrasonication. Next, it was taken out and made a drying treatment at a 120°C-blast drying oven for 0.5 h. Then the filter paper loaded with rGO was immersed in the ZnO precursor solution for 5 min and then kept at 120°C for 0.5 h drying treatment. In order to further increase the loading content of rGO/ZnO, the process of dipping and pulling was repeated three times at the same condition. To research the effect of the proportions of loaded photocatalytic particle on the final photocatalytic activity and antibacterial properties, the prepared multifunctional air filters were named rG, GZ, rGZ1, rGZ2, rGZ3 and rGZ4. Table 1 shows the specific ratio of rGO/ZnO loaded by the prepared air filters. Lastly, the rGO/ZnO/s-GF air filters were obtained by annealing at 400°C for 0.5 h with a heating rate of 5°C min$^{-1}$.

## 2.3. Characterization

In order to study the coating morphology of loaded rGO/ZnO photocatalysis particles on the surface of s-GF, the morphology and element distribution of the multifunctional rGO/ZnO/s-GF air filters were

**Table 1.** Samples of rGO/ZnO/s-GF air filters content with different loading ratios.

| content | name | | | | | |
|---|---|---|---|---|---|---|
| | GZ | rG | rGZ1 | rGZ2 | rGZ3 | rGZ4 |
| ZnO (mol l$^{-1}$) | 0.8 | 0 | 0.8 | 0.8 | 0.8 | 1.6 |
| rGO (ml) | 0 | 2 | 1 | 2 | 3 | 2 |

characterized by scanning electron microscopy (SEM, Zeiss Merlin) with an energy dispersive spectroscopy (EDS, Oxford Instruments). In order to explore the photocatalytic activity of the prepared air filter, the absorbance of the rhodamine B (RhB) dye was measured with an UV–Vis–IR spectrophotometer (Agilent Cary 5000) with a scanning range between 400 and 700 nm. The chemical composition of the multifunctional air filter surface was analysed by Raman spectroscopy (laser 778 nm excitation) and X-ray photoelectron spectroscopy (Thermo ESCALAB 250Xi). The photoluminescence (PL) emission spectra of the samples were collected at 360–660 nm by using RF600 (SHIMADZ), with an excitation wavelength of 325 nm. Room temperature optical absorption of the samples in powders was recorded on a UV3600 (SHIMADZ) with wavelength within 200–800 nm. What is more, the specific surface area changes of the air filter screen loaded with the rGO/ZnO was tested to study the effect of the rGO/ZnO on the enhancement of photocatalytic activity. Brunauer–Emmett–Teller (BET) surface areas were determined by a multipoint BET method using the adsorption data in a relative pressure ($P/P_0$) range of 0.05–0.35 obtained by Gemini VII 5.03 (Micromeritics Instrument Corp.). The electron spin resonance (ESR) spectra which capture the •OH produced by the light time catalyst with 5,5-dimethyl-1-pyrroline N-oxide (DMPO) were collected on a Bruker 300 Paramagnetic spectrometer.

## 2.4. Evaluation of photocatalytic properties of rGO/ZnO/s-GF

In a typical procedure, the rGO/ZnO/s-GF air filter cut into small pieces was dispersed in 50 ml RhB solution of 10 mg l$^{-1}$ [22]. Before lighting, the solution with rGO/ZnO/s-GF was stirred slowly for 0.5 h under shading conditions to reach the adsorption–desorption system equilibrium. Then the solution was exposed to the simulated visible light irradiation produced by a 500 W xenon light source at room temperature. Three microlitres RhB solution was extracted out at regular intervals and centrifuged to remove part of the fallen suspended matter. The dye concentrations of RhB solution during the degradation process were examined by using the UV–Vis absorption spectroscopy.

## 2.5. Antibacterial study

The *E. coli* and *S. aureus* strains were selected as typical microorganisms to detect the antibacterial properties of the s-GF/rGO/ZnO air filter [23,24]. The 60 W LED light was used as the light source and the plate colony counting method was used in this test. The two stains were first suspended in Luria–Bertani (LB) culture medium in a shaking incubator at 37°C for 18 h. Then 50 µl of activated bacteria were dispensed into 5 ml of LB culture medium and cultured for another 12 h under the same conditions. The concentrations of the bacteria were adjusted to $10^5$ colony-forming units (CFU) ml$^{-1}$. These air filters were cut into small pieces and sterilized with 75% alcohol in a clean bench. Subsequently, the dried pieces were added 200 µl of bacteria suspensions and incubated at 37°C under controlled light and dark condition. Afterwards, the pieces were put into a clean tube with 10 ml phosphate buffer saline (PBS) to harvest the bacteria. The bacteria elution was diluted to serial concentrations with a 10-fold gradient. Finally, 100 µl of the diluted bacterial suspensions were coil-coated on LB agar plates and incubated at 37°C for 24 h. The antibacterial activity of the air filters can be obtained by counting colonies and all tests were repeated three times. The antibacterial activity of bacteria growth inhibition rate was evaluated by the following equation:

$$\text{growth inhibition rate } (\%) = \frac{A1 - A2}{A1} \times 100\%,$$

where A1 and A2 are the mean number of bacteria of original s-GF and rGZ2 air filter, respectively.

## 2.6. Filtration performance test of the air filters

The filtration-based method in this study employed a TSI 3160 automated filter tester to determine particle penetration levels as described previously. The TSI 3160 was programmed to measure the per cent penetration of five different size monodisperse particles. Two agglomerated particle counters were used to measure the concentration of the two-way airflow above/below the tested air filters. At the same time, the excited monodisperse aerosol was used to measure the most penetrating particle size (MPPS) of the air filters at a flow rate of 5.3 cm s$^{-1}$. The MPPS and filtration efficiency under different particle sizes is used as describing the critical parameters of filtration performance [25].

# 3. Results

## 3.1. Photocatalytic decolorization of RhB for the photocatalytic efficiency of the rGO/ZnO/s-GF air filters

By decolorizing RhB photosensitive dye in water, the photocatalytic activity of the prepared air filters with different loading content of rGO/ZnO was evaluated. The photocatalytic degradation of RhB solution under visible light using the rGO/ZnO/s-GF air filters was investigated, as shown in figure 2. In order to eliminate the interference of light, the pure RhB solution was tested under visible light. It can be seen that RhB is very stable and the colour of the solution is basically unchanged (figure 2*a*). The result showed the degradation of RhB by the original superfine glass fibre without loading is not intuitively obvious (figure 2*b*). Furthermore, after 2 h of visible light irradiation, the final degradation results of the RhB solution follow the following results: GZ (figure 2*c*) < rGZ1 (figure 2*e*) < rGZ4 (figure 2*g*) < rGZ3 (figure 2*f*) < rGZ2 (figure 2*d*). The colour of RhB solution gradually becomes lighter when different air filters are added, which shows that the photocatalytic efficiency of air filters gradually increases. The above results show that s-GF loaded with rGO/ZnO can improve the photocatalytic activity.

Figure 3 shows the effect of irradiation time on the degradation rate $C/C_0$ of rGO/ZnO/s-GF air filters with different loading concentrations under visible light, wherein $C$ and $C_0$ represents the dye concentration for different light irradiation time and the initial dye concentration, respectively. The degradation rate is equal to 2.4% without any addition in RhB under visible light irradiation. When adding original superfine glass fibre without any loading of rGO/ZnO, the ratio $C/C_0$ increases to 5%, which is considered to be due to the adsorption effect of the original three-dimensional network structure of s-GF. The photocatalytic degradation effects of samples GZ, rGZ1, rGZ2, rGZ3 and rGZ4 were 42%, 70.1%, 95%, 89.2% and 72.1%, respectively. Furthermore, the degradation effect of this quantitative analysis indicates that the air filter after adding rGO/ZnO with a weight ratio of 0.8% rGO has better photocatalytic performance than a pure ZnO-loaded air filter. At the same time, the ratio of rGO/ZnO was also verified to affect the photocatalytic effect of air filters. When the weight ratio of rGO in rGO/ZnO is 1.6%, the photocatalytic degradation efficiency of the air filter reaches 95% on RhB solution.

The above-mentioned could be attributed to the following reasons: the introduced rGO enhances the absorption capacity of dye molecules and promotes the separation of photogenerated electrons and holes in rGO/ZnO and furthermore inhibits the absorption capacity of dye molecules and the ZnO photo-corrosion. However, excessive addition when the weight ratio of rGO in rGO/ZnO is 2.4% will result in increasing the opacity and light scattering to reduce the photocatalytic degradation effect of the filter. The PL emission spectra of rGZ1, rGZ2, rGZ3, rGZ4 air filters are shown in electronic supplementary material, figure S1. The weaker emission peak indicates a greater separation rate of photogenerated electron–hole pairs in the material, which further reflects the stronger photocatalytic activity of the rGZ2. Thus, the best synergetic effect between ZnO and rGO sheets in degrading RhB is attained in air filter rGZ2. In summary, based on the best photocatalytic effect of rGZ2, the subsequent morphology and antibacterial characterization and photocatalytic mechanism analysis were carried out by adopting rGZ2 as the classical sample.

## 3.2. Electron microscopy studies of air filters for morphology and elemental distribution

The morphology of the s-GF air filter before and after loading rGO/ZnO is shown in figure 4. As can be seen in figure 4*a*, the surface of the s-GF without any loading treatment is very smooth and presents a

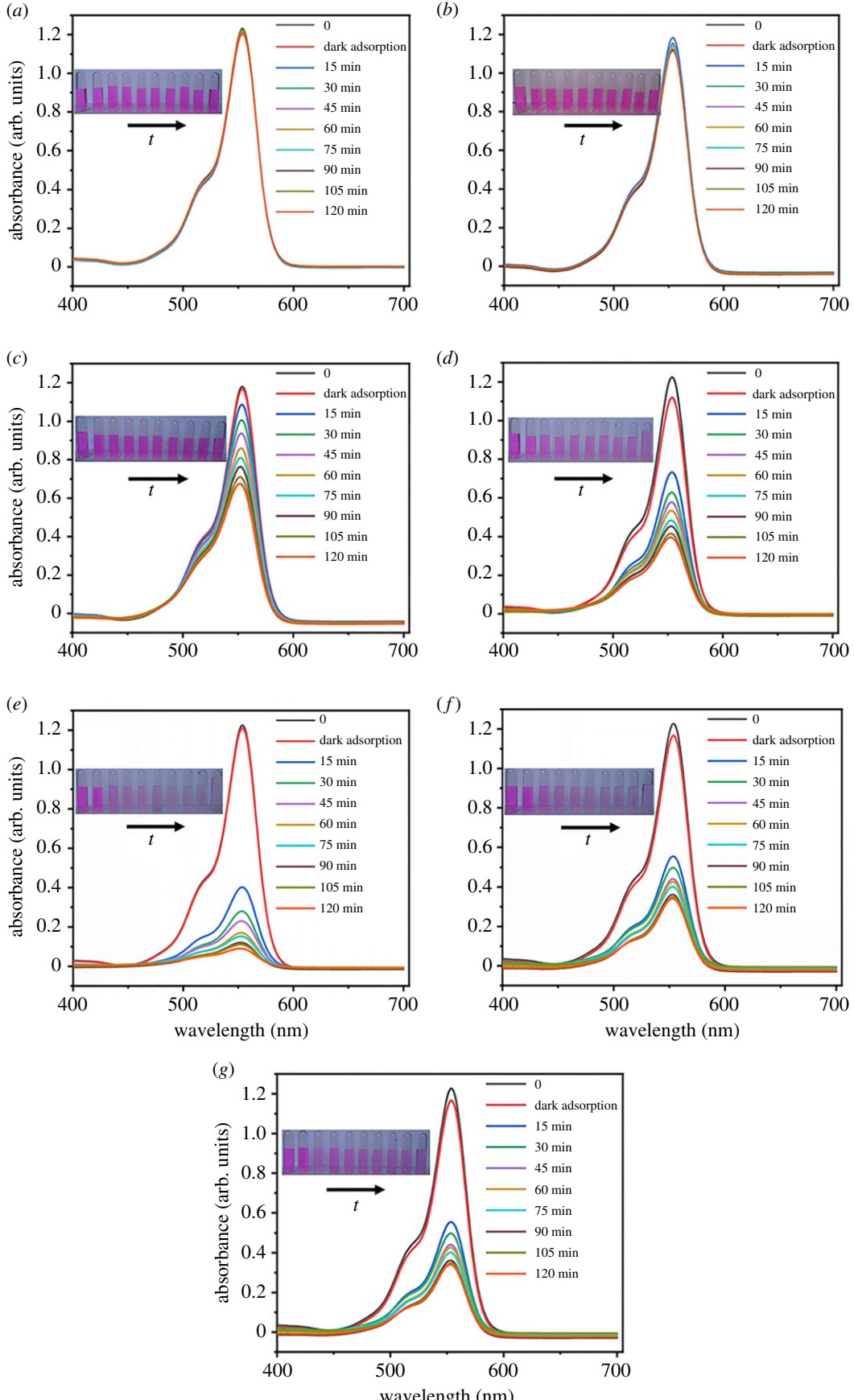

**Figure 2.** Variation of RhB concentration as a function of irradiation time. (*a*) Black control, (*b*) s-GF, (*c*) GZ, (*d*) rGZ1, (*e*) rGZ2, (*f*) rGZ3, (*g*) rGZ4.

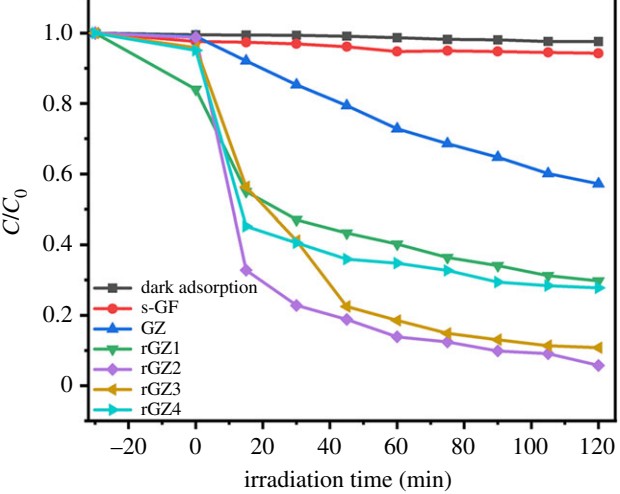

**Figure 3.** Photocatalytic degradation of RhB by air filter.

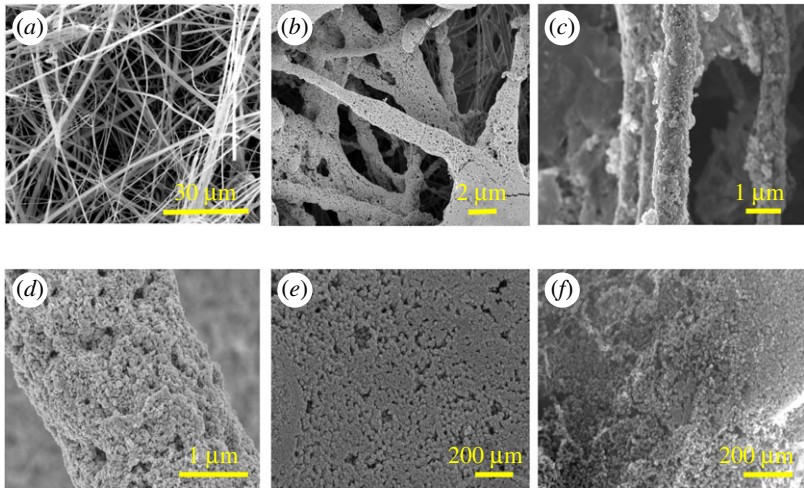

**Figure 4.** Scanning electron microscope (SEM) images of (*a*) s-GF, (*b*) GZ, (*c*) rGZ2, (*d,e*) enlarged image of GZ, (*f*) enlarged image of rGZ2.

three-dimensional network structure, which can well explain the cause of excellent air filtration performance of the original s-GF. After immersing and annealing in the ZnO precursor solution, the fibre surface was tightly and uniformly wrapped by ZnO, shown in figure 4*b*. After adsorbing a certain amount of rGO seed layer on the fibre surface, the *in situ* growth based on sol–gel process of ZnO is as shown in the figure 4*c*. It can be found that the dispersion of ZnO on rGO seed layer is better than that without rGO addition and the three-dimensional network structure of the s-GF air filter is still strong and stable without damage. Further, a magnified observation of a single s-GF-loaded ZnO is as shown in figure 4*d*, presenting that the ZnO are loaded relatively uniformly on the surface of s-GF. However, regardless of whether attaching the rGO seed layer, the agglomeration of a small number of ZnO was found, and this phenomenon was more obvious at the crossing of the three-dimensional network structure of s-GF. The morphology of nanoparticles on the surface of GZ air filters was enlarged as shown in figure 4*e*, it was found that the average diameter of nanoparticles was between 50 and 100 nm and distributed relatively uniformly. Compared with figure 4*e*, figure 4*f* shows the average size of ZnO reaching to the range of 20–50 nm when the rGO seed layer is loaded on the surface of s-GF, indicating that ZnO can not only improve the agglomeration phenomenon between graphene sheets [17], but also found that rGo can improve the dispersibility of ZnO particles due to the formation of the rGO/ZnO composite structure. In general, a special three-dimensional network structure composed of three layers of s-GF, rGO and ZnO was prepared for air filter. At the same time, the morphological analysis also proves the tight coating of rGO/ZnO.

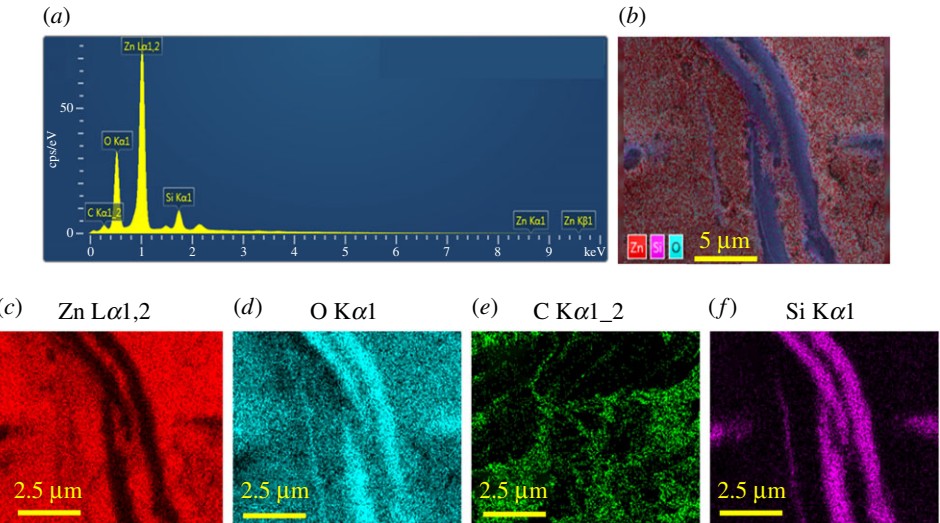

**Figure 5.** EDS of rGZ2 air filter (*a,b*). EDX elemental map, (*c*) Zn, (*d*) O, (*e*) C, (*f*) Si.

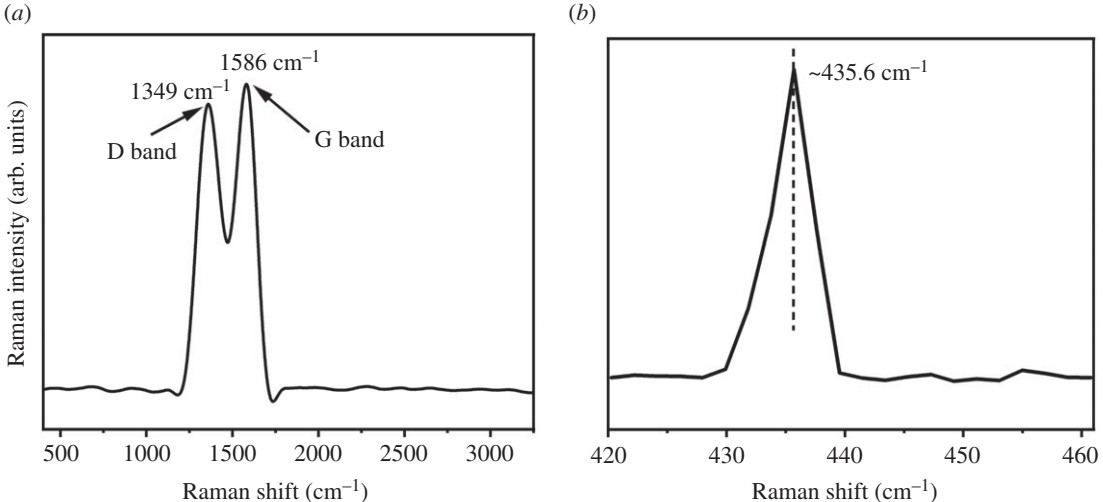

**Figure 6.** Raman spectrum of (*a*) rGZ2 and (*b*) partial magnification 420–460 nm.

## 3.3. EDS and Raman spectroscopy research of rGZ2 air filter for element distribution and successful loading of the rGO/ZnO

The mapping and element distribution of the rGO/ZnO/s-GF air filter is shown in figure 5. Among them, the specific Zn, O, Si and C elemental distribution of rGZ2 sample is shown in figure 5*a,b* indicating that the detailed content corresponding to Zn (32.16%), O (46.72%), C (14.40%) and Si (6.72%) was detected. Furthermore, as shown in the mapping in figure 5*c–f*, obviously the presence of Si element belongs to the base matrix material of s-GF, which can explain why O element content in the element distribution of rGZ2 air filter is slightly more than Zn element. All the above-mentioned proves that the rGO/ZnO is uniformly distributed on the s-GF surface.

The Raman spectrum was analysed to obtain the material composition on the surface of the multifunctional rGZ2 air filters. The Raman spectra of rGZ2 sample is shown in figure 6*a*. The Raman peaks at 1349 and 1586 cm$^{-1}$ correspond to the defect-induced D band and characteristic G band of rGO. The D band represents sp3 carbon defects and the G band corresponds to an ordered sp2 carbon network [26–28]. This also further proves that rGO is attached to the substrate of the superfine glass fibre. The Raman peak in the multifunctional rGO/ZnO/s-GF air filters can be observed at 435.6 cm$^{-1}$ due to the presence of ZnO nanoparticles [29] in figure 6*b*. This is also a good proof that s-GF first loads rGO and then provides ZnO with more growth sites to obtain a special three-dimensional network of air filters based on a rGO/ZnO/s-GF multilayer.

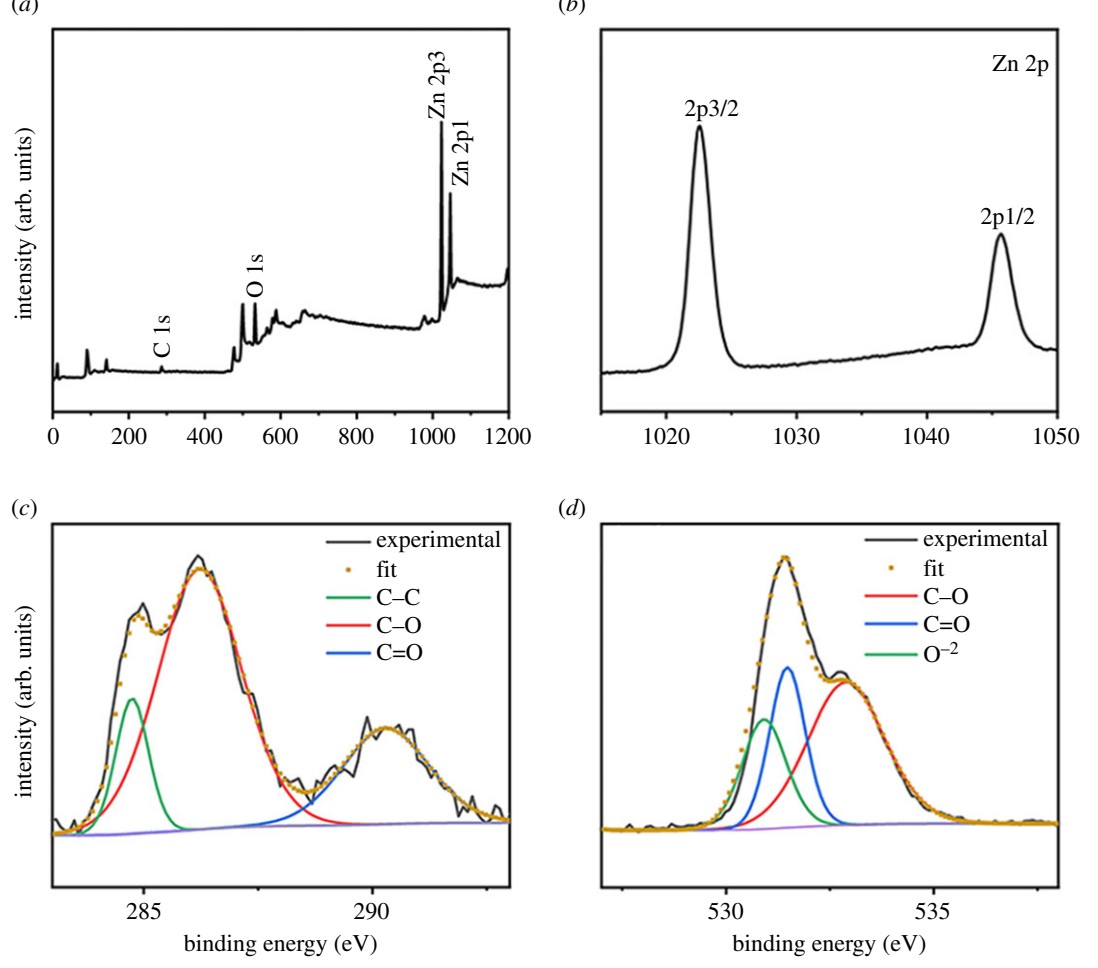

**Figure 7.** X-ray photoelectron spectroscopy spectra of rGZ2. (*a*) survey, (*b*) Zn 2p, (*c*) C 1s, (*d*) O 1s.

## 3.4. X-ray photoelectron spectroscopy analysis of rGZ2 air filter for the surface composition and chemical states

The chemical composition and element state on the surface of the rGZ2 air filter was characterized by X-ray photoelectron spectroscopy (XPS). The full scanning spectrum shows that the prepared rGZ2 air filter only contains the three elements Zn, C and O and no other elemental peak can be found, as shown in figure 7*a*. Two different characteristic peaks of Zn 2p appear at 1022.8 and 1045.4 eV, respectively, which correlate with 2p3/2 and 2p1/2 states of ZnO nanoparticles in figure 7*b* and further clarify the existence of $Zn^{2+}$ in ZnO [30]. Furthermore, the high-resolution scan of C 1s spectra (figure 7*c*) shows three different peaks at 284.8, 286.2 and 290.2 eV, which are designated as C–C, C–O and C=O bonds of the air filter, respectively [31,32]. Figure 7*d* shows the spectrum of O 1s. These fitted peaks are at 530.2, 531.5 and 532.85 eV, corresponding to $O^{2-}$ ions, Zn–O bond in the ZnO lattice and oxygen-containing groups (C–OH, C–O–C) [16], which further illustrates that the surface of the rGZ2 air filter is loaded with rGO and ZnO.

## 3.5. Photoluminescence spectrum analysis and UV-absorption characteristics

Figure 8*a* shows the photoluminescence (PL) emission (excited at 325 nm) spectra of the GZ and rGZ2 air filters. A strong ultraviolet emission peak at 386 nm corresponding to near band-edge emission is attributed to the excitonic recombination of a hole in the valence band and an electron in the conduction band. The intensity of the PL emission band can be used to reflect the lifetime of electron–hole pairs because PL comes from the recombination of electron–hole pairs after the photocatalyst is irradiated with light [33]. Electrons excited by light on zinc oxide can be transported to the interface of rGO, which effectively reduces the recombination of photogenerated electron–hole pairs and thus

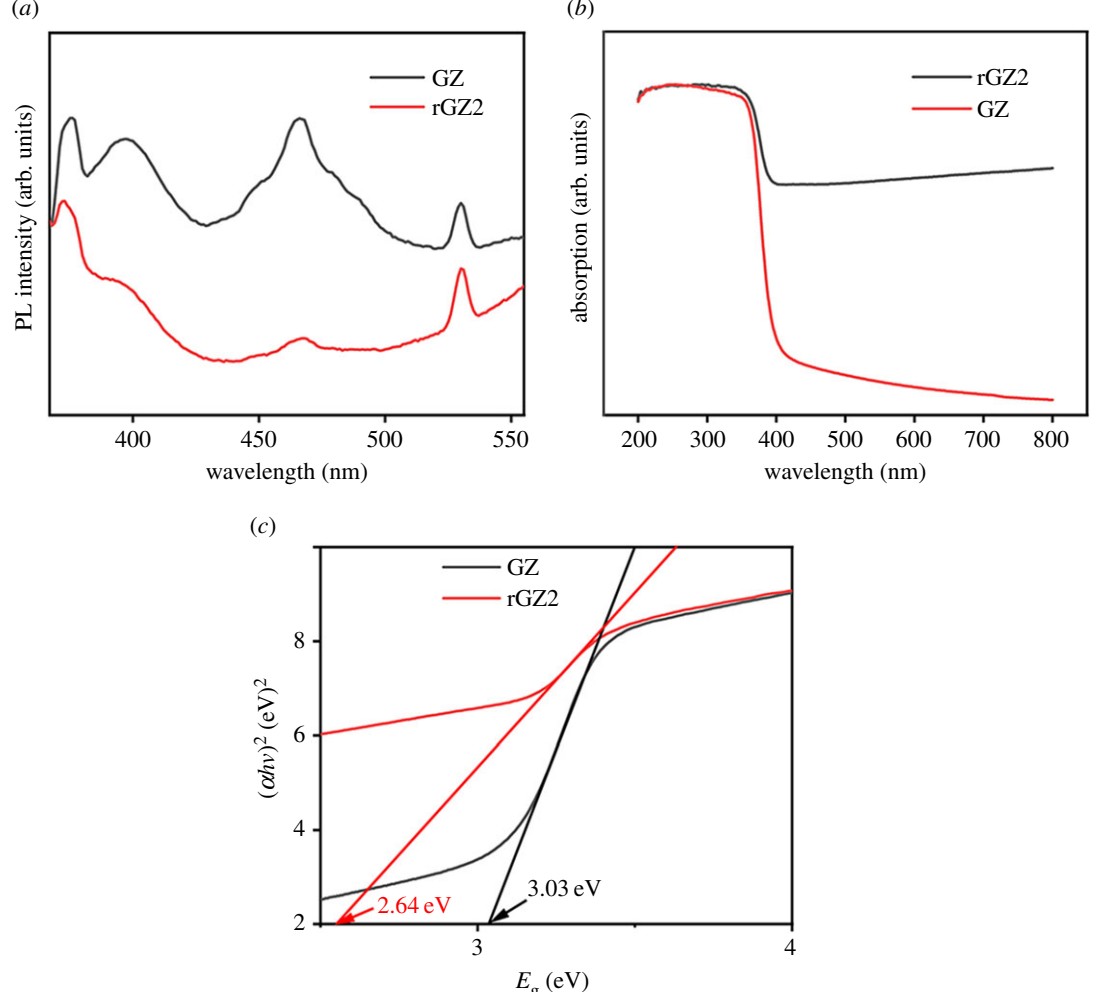

**Figure 8.** The room temperature PL emission spectra of the GZ and rGZ2 are shown in (*a*). UV-DRS analysis for the rGZ2 and GZ air filters (*b*). A plot of $(\alpha h v)^2$ versus the band gap energy (eV) for GZ and rGZ2 filters (*c*).

weakens the emission peak of the rGZ2 [34,35]. The weaker emission peak indicates a greater separation rate of photogenerated electron–hole pairs in the material, which further reflects the stronger photocatalytic activity of the rGZ2 air filter.

Figure 8*b* depicts the UV-absorption spectra of GZ and rGZ air filters recorded in the range of 200–800 nm. It can be seen that the GZ only targets the absorption of ultraviolet light with a wavelength below 380 nm. Different from GZ, the light absorption range of rGZ2 air filter paper is obviously red-shifted, showing strong absorption ability in the visible light region. The band gap energies of GZ and rGZ2 calculated from Tauc's equation: $(\alpha h v)^2 = A(h v - E_g)^2$ [36] are shown in figure 8*c*. It can be seen the rGO/ZnO shows extended absorption as compared to only ZnO due to the cooperative interface interaction between ZnO and rGO [37,38]. This means that all photon energy of visible light can be absorbed by GZ, so the introduction of rGO in GZ can greatly expand the light absorption range and improve the photocatalytic activity of the air filter. The spectroscopic results of PL and UV-DRS indicate that the photocatalytic activity of rGZ2 should indeed be higher than that of the GZ air filter.

## 3.6. The photocatalytic reaction mechanism of the rGO/ZnO/s-GF air filter

In order to further reveal the active groups during the photocatalytic degradation and antibacterial process of rGZ2, the ESR spectra of the free radicals captured by DMPO in rGZ2 were studied. As shown in figure 9, after 4, 8 and 12 min of irradiation, the six characteristic peaks of DMPO−O$_2^{\bullet-}$ (figure 9*b*) and four characteristic peaks of DMPO-•OH (figure 9*c*) can be clearly seen in rGZ2 air filter [39,40]. By contrast, rGZ2 in the dark did not find a significant signal peak of free radicals, indicating that the air filter did not generate free radicals in the absence of light. This reveals that the

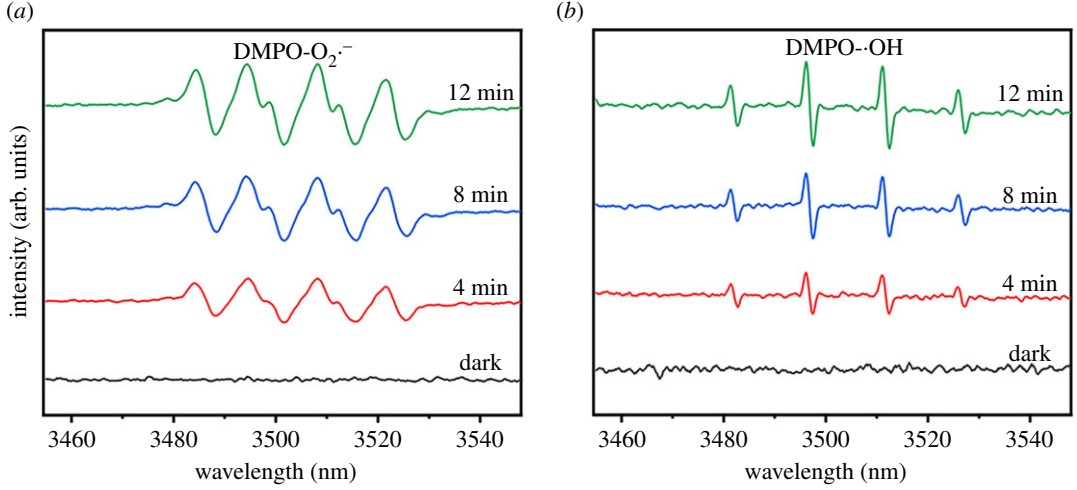

**Figure 9.** ESR spectra of (*a*) $O_2^{\bullet-}$ and (*b*) •OH obtained from the rGZ2 air filter after different exposure times under simulated light.

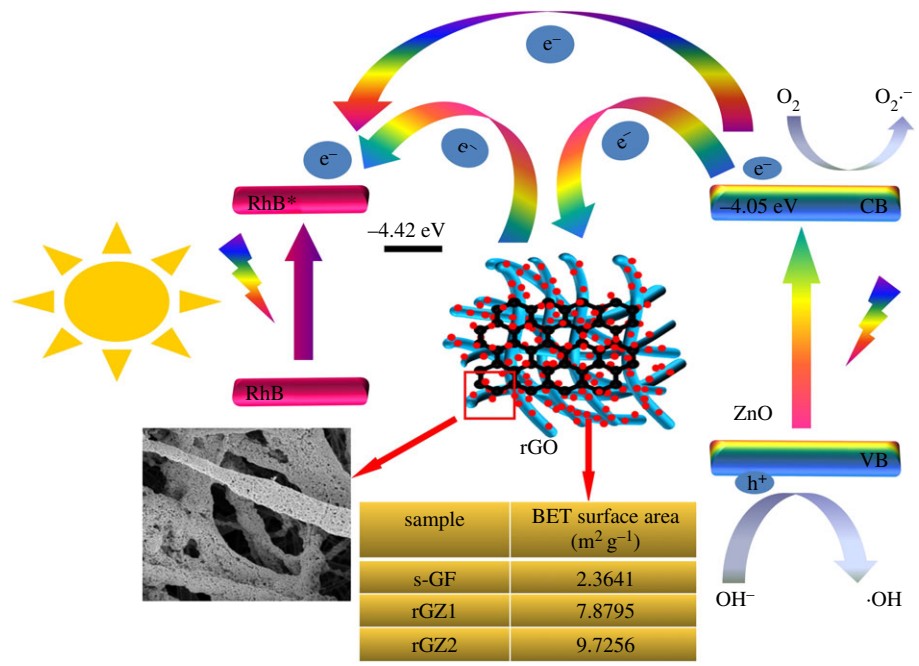

**Figure 10.** Schematic diagram of photocatalytic degradation principle.

rGZ2 air filter produces two free radicals DMPO-•OH and DMPO$-O_2^{\bullet-}$ under light conditions, and this also plays an important role in the research on the mechanism of photocatalytic degradation of organic dyes and rapid sterilization. The photocatalytic reaction mechanism of the rGO/ZnO/s-GF air filter is shown in figure 10. One side, the change between the s-GF air filter before and after loading rGO/ZnO was studied. The specific surface area of the samples, s-GF, rGZ1 and rGZ2, measured by the BET method, are 2.3641, 7.8795 and 9.7256 $m^2 \, g^{-1}$, respectively. This proves that the specific surface area may also play a very important role in the improvement of photocatalytic performance, because it can better capture microorganisms in the air to provide more photocatalytic active sites. On the other hand, the photogenerated electrons move from the valence band to the conduction band of ZnO under the irradiation of visible light. As an intermediate carrier, rGO seed layer accepts photogenerated electrons excited from the conduction band of ZnO, which greatly improves the utilization efficiency of photogenerated electron pairs. These excited photogenerated electrons can capture dissolved oxygen molecules and generate ROS, which will further decompose the RhB dye mainly due to the different energy level between rGO and ZnO. The work function value of rGO is −4.42 eV, which is lower than the conduction band value of −4.05 eV of ZnO [41,42]. Therefore, in the

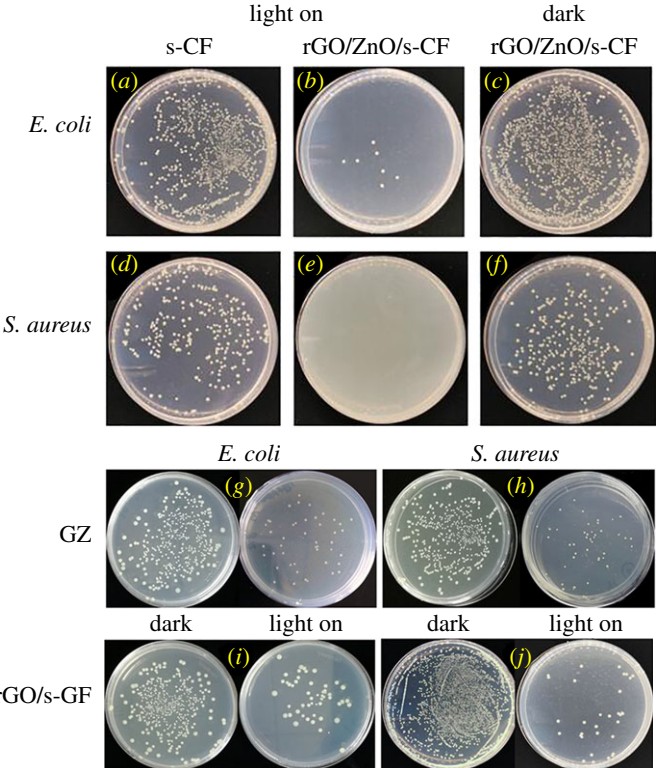

**Figure 11.** Images of antibacterial activities of (*a*) cultivation of *E. coli* on s-GF and (*b*) rGZ2 under light conditions, (*c*) rGZ2 in dark; (*d*) s-GF and (*e*) rGZ2 against *S. aureus* under light conditions, (*f*) rGZ2 in dark. Images of antibacterial activities of (*g,h*) the GZ and (*i,j*) rGO/s-GF air filter.

presence of rGO of the multifunctional air filters, it can achieve effective separation of charges to complete superior photocatalytic performance under visible light [32,43]. The two above-mentioned reasons prove that the rGO/ZnO/s-GF air filters prepared have good photocatalytic activity, which provides more possibilities for further industrial production in crowded places where the multifunctional air filter with excellent filtration and antibacterial are required [44–46].

## 3.7. Photocatalytic antibacterial performance of the multifunctional antibacterial air filter

The bactericidal performance of the rGZ2 air filter was evaluated by assessing the number of colonies formed on the culture plate [47]. In order to explore the effect of light, as shown in figure 11, the antibacterial experiments of air filters under light and dark conditions were tested. The filter without photocatalyst was cultured for 4 h under simulated visible light irradiation (figure 11*a*) and its average colony count was 780 CFU ml$^{-1}$. The average number of colonies obtained by growing *in situ* loaded rGO/ZnO photocatalyst filter (figure 11*b*) under the same conditions is 7 CFU ml$^{-1}$. It can be seen that the rGZ2 air filter has a good bactericidal effect, but it cannot rule out the influence of light on the experiment for the time being, so the rGZ2 air filter sample was cultured under shading conditions (figure 11*c*), the average number of colonies is 804 CFU ml$^{-1}$. This shows that the multifunctional air filter has a very excellent photocatalytic effect by visible light. At the same time, *S. aureus* was also cultured under the same conditions and the average colony counts were 324 CFU ml$^{-1}$ (figure 11*d*), 0 CFU ml$^{-1}$ (figure 11*e*) and 309 CFU ml$^{-1}$ (figure 11*f*). This result also verified the excellent photocatalytic sterilization effect of this rGZ2 air filter under visible light. At the same time, in order to prove the synergistic enhancement effect of rGO and ZnO in the rGZ2 sample, the antibacterial experiment of GZ and rG was also tested and the results are shown in figure 11*h–j*. From the results, the antibacterial activity of GZ and rG air filters against *E. coli* and *S. aureus* can also reach 80%. The antibacterial activity of ZnO and rGO has been confirmed [48], but the antibacterial effect of these two kinds of filters cannot reach 99% of rGZ2's bactericidal effect. This result also shows the superiority of the prepared rGZ2 air filter material.

Generally, the rGO/ZnO loaded on the surface of s-GF is not toxic to bacterial cells and only when light is stimulated to produce photogenerated electron–holes, the bacteria can be inactivated.

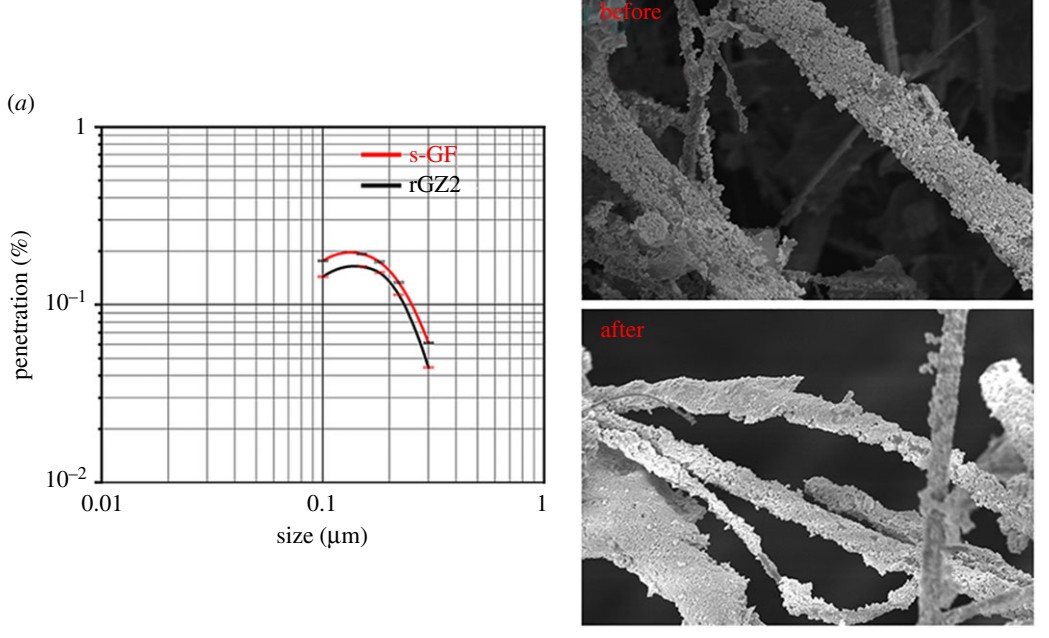

**Figure 12.** The fractional efficiency filter test of the s-GF and rGZ2 air filter (*a*) and the morphology of the sample under the scanning electron microscope before and after centrifugation (*b*).

Photocatalytic sterilization generally includes two mechanisms of action: direct and indirect sterilization [49,50]. Direct sterilization mainly refers to the photogenerated $e^-$ and photogenerated $h^+$ produced by the rGO/ZnO directly attacking microbial cells, causing the cell wall, cell membrane and intracellular substances to be destroyed and lost activity; the indirect sterilizer is the reaction of photogenerated $e^-$ or photogenerated $h^+$ with $H_2O$ or dissolved oxygen to generate reactive oxygen species such as $O_2^-$, •OH and $H_2O_2$. These reactive oxygen species act on bacterial cells and cause cell death [51–54]. The free radicals generated in the photocatalysis process have also been confirmed by ESR spectra. In short, the multifunctional air filter has a reliable antibacterial effect whether it is directly or indirectly under the visible light. This further proves the advantages of the prepared rGZ2 air filters in hospitals and other places that require a high degree of cleanliness.

## 3.8. Filter performance and structural stability

By comparing the filtration efficiency of the s-GF air filters with the one without any addition, the filtration performance of the synthesized multifunctional filters is evaluated. It can be seen from the figure 12*a* that whether the surface of s-GF is loaded with ZnO/rGO-based particles, the filtration efficiency remains at 99.9% under the test of five kinds of particles with different diameters. The MPPS of the s-GF and the rGZ2 air filter is 0.12216 and 0.12104 µm, respectively. This result indicates that the loading of composite photocatalytic particles on the surface of glass microfibre does not reduce the filtration performance of the air filter. The air filtration experiments direct that the rGZ2 air filters still maintain a stable and effective interception for aerosol particles. After immersing rGZ2 in an aqueous solution and centrifuging at a speed of 12 000 r.p.m. for 20 min, the SEM morphology changes of rGZ2 were studied, as shown in figure 12*b*. In order to verify the loading stability of rGO/ZnO photocatalytic materials on s-GF and the air filter, structural stability is in practical application. The results show that rGO/ZnO still maintains a tight package of s-GF, and the three-dimensional network structure is also intact. This indicates the tightness of rGO/ZnO load on s-GF and the good structural stability of rGZ2 air filter [25,55].

## 4. Conclusion

In summary, a multifunctional superfine glass fibre air filter with fast photocatalytic response and antibacterial properties under visible light by loading rGO/ZnO was successfully synthesized. The method loaded rGO/ZnO on the surface of s-GF by *in situ* sol–gel process followed by calcination,

the subsequent characterization of the morphology and structure of the air filter confirmed the special three-dimensional network structure of the filters and the tight loading and uniform distribution of rGO/ZnO. The specific surface area of the rGO/ZnO/s-GF air filter before and after loading rGO/ZnO increased from 2.3641 to 9.7256 $m^2\,g^{-1}$. Also adding rGO to the s-GF can effectively improve the photocatalytic activity of ZnO and increase the use of visible light. Under the visible light irradiation of a 500 W xenon lamp, the degradation rate of the rGZ2 air filter to RhB within 2 h is 95%, which is twice the degradation rate of 42% of the pure GZ air filter. Besides, the antibacterial test of *E. coli* and *S. aureus* was performed on the rGZ2 air filter within 4 h. Under 60 W LED simulated visible light, the inactivation rate of *E. coli* reached 99.9% and the inactivation rate of *S. aureus* up to 100%. The PL and DRS measurement results confirm that the compound of rGZ2 can effectively suppress recombination of photogenerated electrons and holes. The actual filtration efficiency of the air filter is 99.9%, rGZ2 still maintains good structural stability and load tightness after high-speed centrifugation. At the same time making some inferences about the photocatalytic degradation and antibacterial mechanism proved the principle of the rGZ2 air filter improving photocatalytic degradation in a short time under visible light through the generation of •OH and $O_2^{\bullet-}$ radicals in the ESR test. This further shows that the multifunctional air filter can be used for rapid response air purification and antibacterial treatment under visible light conditions.

Ethics. This study does not present research with ethical considerations.

Data accessibility. Material characterization results and methylene blue aqueous solution degradation data are available at the Dryad Digital Repository: https://doi.org/10.5061/dryad.8931zcrpr [56].

The data are provided in the electronic supplementary material [57].

Authors' contributions. Conceptualization, F.Z. and L.L.; methodology, Y.L.; validation, Y.L., F.Z. and Y.Z.; formal analysis, Y.L., Z.C. and Y.Z.; software, Y.L.; investigation, Y.L. and G.F.; resources, F.Z., M.D., D.Q. and J.Y.; data curation, Y.L.; writing—original draft preparation, Y.L.; writing—review and editing, Y.L and F.Z.; visualization, F.Z and Y.L.; supervision, Y.L.; project administration, F.Z and L.L.; funding acquisition, F.Z. and L.L. All authors have read and agreed to the published version of the manuscript.

Competing interests. We have no competing interests.

Funding. This research was funded by Ministry of Industry and Information Technology of The People's Republic of China, grant no. 2019-00900-1-1; Research Program of Chongqing Municipal Education Commission, grant nos. KJQN201901348, KJCX2020048; Chongqing University of Arts and Sciences, grant no. R2019FXC06; Chongqing Science and Technology Bureau, grant nos. T04040012 and cstc2019jcyjjqx0021.

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
