## [Peer Review File · Royal Society Open Science]

Review History

RSOS-202285.R0 (Original submission)

Review form: Reviewer 1

Is the manuscript scientifically sound in its present form?

Yes

Are the interpretations and conclusions justified by the results?

Yes

Is the language acceptable?

Yes

Do you have any ethical concerns with this paper?

No

Have you any concerns about statistical analyses in this paper?

No

Recommendation?

Accept with minor revision (please list in comments)

Comments to the Author(s)

- (1) The detailed calculation of surface area of the samples-GF, rGZ1 and rGZ2 measured by the BET method should be explained;
- (2) The adhesion of rGZ and ZnO on glass fiber need be provided.

Review form: Reviewer 2**Is the manuscript scientifically sound in its present form?**

No

Are the interpretations and conclusions justified by the results?

No

Is the language acceptable?

No

Do you have any ethical concerns with this paper?

No

Have you any concerns about statistical analyses in this paper?

No

Recommendation?

Major revision is needed (please make suggestions in comments)

Comments to the Author(s)

A superfine glass fiber air filter with rapid response to photocatalytic antibacterial properties under visible light by loading rGO/ZnO

Manuscript No: RSOS-202285

I reviewed the article entitled, "A superfine glass fiber air filter with rapid response to Photocatalytic antibacterial properties under visible light by loading rGO/ZnO. The manuscript reports the preparation of superfine glass fiber/GO/ZnO composite materials with photocatalytic/antibacterial activity. The idea of making a photocatalytic air filter that can simultaneously filter/photo-kill microbes is a good one but the scientific discussion is rather poor and there are important points that need to be addressed to improve the scientific value of the manuscript. For instance,

- 1- No air-filtration experiments were performed as per the title of the manuscript.
- 2- Figure 8: An arbitrary mechanism is proposed without providing experimental evidences of the data included in Fig. 8.
- 3- The reasons for difference in photocatalytic/antibacterial activity are not discussed with adequate justification and assigning the higher activity of rGZ2 to surface area effect (and why it has high surface area?) is not sufficient. Other claim that rGO reduced the e⁻-h⁺ recombination and increase absorption in visible region need experimental evidence (PL measurements, and diffuse reflectance measurement (band gap evaluation). That rGO enhances dyes adsorption

needs to be justified with experimental evidence. That rGO leads to a decrease in ZnO particle size may be explained better providing PSD histograms in ESI.

4- The experimental procedure for the synthesis of ZnO is sol-gel processing followed by calcination (400 °C) and cannot be called “in-situ pyrolysis growth at low temperature” as Pyrolysis involves the thermal decomposition of materials at elevated temperatures in an inert atmosphere and 400 °C is not that low. Moreover, the Material and Method section reports the use of Zinc acetate dehydrate (material) and then anhydrous Zn-acetate (experimental).

5- Relevant experimental details (such as charge correction in XPS etc) are missing

6- The antibacterial tests are reported to be performed under LED illumination (see the conclusion part) but there is no mention of it in the experimental part which actually reports the use of 500 W Xenon as light source in the photocatalytic tests.

7- The theoretical fit on experimental XPS data (or fitting residue) is missing, making the fitting less reliable. The assignment of 283.36 eV photoemission peak to C-O-Zn needs reevaluation as the cited references (30, 31) say nothing about it.

8- The claim that rGO/ZnO loaded on the surface of s-GF is not toxic to bacterial cells is not totally true. ZnO may show some antibacterial properties in dark (doi: 10.1016/j.jiec.2016.10.013) but no such effects are seen in Fig. 9.

9- Finally, the English of the manuscript require proof reading as there are dozens of words with no spacing in between and other grammatical error.

Review form: Reviewer 3

Is the manuscript scientifically sound in its present form?

No

Are the interpretations and conclusions justified by the results?

No

Is the language acceptable?

Yes

Do you have any ethical concerns with this paper?

No

Have you any concerns about statistical analyses in this paper?

No

Recommendation?

Major revision is needed (please make suggestions in comments)

Comments to the Author(s)

Comments

This manuscript reports a rGO/ZnO modified superfine glass fiber air filter by dipping and in-situ pyrolysis growth. The photocatalytic and antibacterial properties of the rGO/ZnO/s-GF air filter was evaluated. However, the author did not give the filtration performance of the air filter.

Besides, the characterization and data analyses in this paper are too simple and insufficient. The paper is not very well arranged, and may not be accepted in the current form.

1. In summary section, the author stated the importance and challenges of high-performance air filter, and this article also focuses on superfine glass fiber air filters, but there is no air purification performance data in the article.
2. Why does rGZ2 air filter have the best photocatalytic RhB decolorization performance? The content of ZnO and rGO is not the highest, and the author should give a clear explanation.
3. What was the author use in section 3.1 and 3.2, anhydrous zinc acetate or zinc acetate dihydrate?
4. In section 4.2, "As can be seen in Figure 4a, the surface of the s-GF without any loading treatment is very smooth and presents a three-dimensional network structure, which can well explain the cause of excellent air filtration performance of the original s-GF." This part lack of data support.
5. In introduction section, "Furthermore, adding ZnO nanoparticles can also inhibit the agglomeration between rGO layered structures to a certain extent." However, it is pointed out that the addition of rGO can solve the agglomeration of ZnO. (in section 4.2). It is contradictory.
6. It is reported that GO also has antibacterial performance. The author should give the antibacterial performance of the rGO/s-GF air filter to better explain the sterilization mechanism of the filters.
7. Typos and format errors in the manuscript should be corrected. n, e.g., in section 3.2, "... distilled waterand then 30mL...", "... with a heating rate of5°C/min", missing spaces.

Decision letter (RSOS-202285.R0)

Dear Dr Zhai

The Editors assigned to your paper RSOS-202285 "A superfine glass fiber air filter with rapid response to photocatalytic antibacterial properties undervisible light by loading rGO/ZnO" have now received comments from reviewers and would like you to revise the paper in accordance with the reviewer comments and any comments from the Editors. Please note this decision does not guarantee eventual acceptance.

Please submit your revised manuscript and required files (see below) no later than 21 days from today's (ie 09-Jun-2021) date. Note: the ScholarOne system will 'lock' if submission of the revision is attempted 21 or more days after the deadline. If you do not think you will be able to meet this deadline please contact the editorial office immediately.

on behalf of R. Kerry Rowe (Subject Editor)
openscience@royalsociety.org

Associate Editor Comments to Author:

Comments to the Author:

Three reviewers have reported on your work. Given the concerns that reviewers 2 (who has supplied a document to review) and 3 have regarding your work, we cannot consider the paper suitable for publication until and unless the comments of the reviewers are satisfactorily addressed. Please make sure that your revision takes the reviewers' concerns into account and that you provide a full point-by-point response outlining the changes made. The revision will be sent back to reviewers 2 and 3 for further consideration.

Reviewer comments to Author:

Reviewer: 1

Comments to the Author(s)

- (1) The detailed calculation of surface area of the samples-GF, rGZ1 and rGZ2 measured by the BET method should be explained;
- (2) The adhesion of rGZ and ZnO on glass fiber need be provided.

Reviewer: 2

Comments to the Author(s)

A superfine glass fiber air filter with rapid response to photocatalytic antibacterial properties under visible light by loading rGO/ZnO

Manuscript No: RSOS-202285

I reviewed the article entitled, "A superfine glass fiber air filter with rapid response to Photocatalytic antibacterial properties under visible light by loading rGO/ZnO. The manuscript reports the preparation of superfine glass fiber/GO/ZnO composite materials with photocatalytic/antibacterial activity. The idea of making a photocatalytic air filter that can simultaneously filter/photo-kill microbes is a good one but the scientific discussion is rather poor and there are important points that need to be addressed to improve the scientific value of the manuscript. For instance,

1- No air-filtration experiments were performed as per the title of the manuscript.

2- Figure 8: An arbitrary mechanism is proposed without providing experimental evidences of the data included in Fig. 8.

3- The reasons for difference in photocatalytic/antibacterial activity are not discussed with adequate justification and assigning the higher activity of rGZ2 to surface area effect (and why it has high surface area?) is not sufficient. Other claim that rGO reduced the e⁻-h⁺ recombination and increase absorption in visible region need experimental evidence (PL measurements, and diffuse reflectance measurement (band gap evaluation). That rGO enhances dyes adsorption needs to be justified with experimental evidence. That rGO leads to a decrease in ZnO particle size may be explained better providing PSD histograms in ESI.

4- The experimental procedure for the synthesis of ZnO is sol-gel processing followed by calcination (400 °C) and cannot be called “in-situ pyrolysis growth at low temperature” as Pyrolysis involves the thermal decomposition of materials at elevated temperatures in an inert atmosphere and 400 °C is not that low. Moreover, the Material and Method section reports the use of Zinc acetate dihydrate (material) and then anhydrous Zn-acetate (experimental).

5- Relevant experimental details (such as charge correction in XPS etc) are missing

6- The antibacterial tests are reported to be performed under LED illumination (see the conclusion part) but there is no mention of it in the experimental part which actually reports the use of 500 W Xenon as light source in the photocatalytic tests.

7- The theoretical fit on experimental XPS data (or fitting residue) is missing, making the fitting less reliable. The assignment of 283.36 eV photoemission peak to C-O-Zn needs reevaluation as the cited references (30, 31) say nothing about it.

8- The claim that rGO/ZnO loaded on the surface of s-GF is not toxic to bacterial cells is not totally true. ZnO may show some antibacterial properties in dark (doi: 10.1016/j.jiec.2016.10.013) but no such effects are seen in Fig. 9.

9- Finally, the English of the manuscript require proof reading as there are dozens of words with no spacing in between and other grammatical error.

Reviewer: 3

Comments to the Author(s)

Comments

This manuscript reports a rGO/ZnO modified superfine glass fiber air filter by dipping and in-situ pyrolysis growth. The photocatalytic and antibacterial properties of the rGO/ZnO/s-GF air filter was evaluated. However, the author did not give the filtration performance of the air filter. Besides, the characterization and data analyses in this paper are too simple and insufficient. The paper is not very well arranged, and may not be accepted in the current form.

1. In summary section, the author stated the importance and challenges of high-performance air filter, and this article also focuses on superfine glass fiber air filters, but there is no air purification performance data in the article.

2. Why does rGZ2 air filter have the best photocatalytic RhB decolorization performance? The content of ZnO and rGO is not the highest, and the author should give a clear explanation.

3. What was the author use in section 3.1 and 3.2, anhydrous zinc acetate or zinc acetate dihydrate?

4. In section 4.2, “As can be seen in Figure 4a, the surface of the s-GF without any loading treatment is very smooth and presents a three-dimensional network structure, which can well explain the cause of excellent air filtration performance of the original s-GF.” This part lack of data support.

5. In introduction section, "Furthermore, adding ZnO nanoparticles can also inhibit the agglomeration between rGO layered structures to a certain extent." However, it is pointed out that the addition of rGO can solve the agglomeration of ZnO. (in section 4.2). It is contradictory.
6. It is reported that GO also has antibacterial performance. The author should give the antibacterial performance of the rGO/s-GF air filter to better explain the sterilization mechanism of the filters.
7. Typos and format errors in the manuscript should be corrected. n, e.g., in section 3.2, "... distilled waterand then 30mL...", "... with a heating rate of5°C/min", missing spaces.

===PREPARING YOUR MANUSCRIPT===

===PREPARING YOUR REVISION IN SCHOLARONE===

Please ensure that you include a summary of your paper at Step 2 'Type, Title, & Abstract'. This should be no more than 100 words to explain to a non-scientific audience the key findings of your

research. This will be included in a weekly highlights email circulated by the Royal Society press office to national UK, international, and scientific news outlets to promote your work.

<https://royalsociety.org/journals/authors/author-guidelines/#supplementary-material> to include a suitable title and informative caption. An example of appropriate titling and captioning may be found at https://figshare.com/articles/Table_S2_from_Is_there_a_trade-off_between_peak_performance_and_performance_breadth_across_temperatures_for_aerobic_sc_ope_in_teleost_fishes_/3843624.

Author's Response to Decision Letter for (RSOS-202285.R0)

See Appendix A.

RSOS-202285.R1 (Revision)

Review form: Reviewer 1

Is the manuscript scientifically sound in its present form?

Yes

Are the interpretations and conclusions justified by the results?

Yes

Is the language acceptable?

Yes

Do you have any ethical concerns with this paper?

No

Have you any concerns about statistical analyses in this paper?

No

Recommendation?

Accept as is

Comments to the Author(s)

The authors have modified the manuscript according to the reviewers' comments, which can be accepted directly.

Review form: Reviewer 2

Is the manuscript scientifically sound in its present form?

Yes

Are the interpretations and conclusions justified by the results?

Yes

Is the language acceptable?

Yes

Do you have any ethical concerns with this paper?

No

Have you any concerns about statistical analyses in this paper?

No

Recommendation?

Accept with minor revision (please list in comments)

Comments to the Author(s)

A superfine glass fiber air filter with rapid response to photocatalytic antibacterial properties under visible light by

loading rGO/ZnO

Manuscript No: RSOS-202285-R1

I reviewed the revised manuscript article entitled, "A superfine glass fiber air filter with rapid response to Photocatalytic antibacterial properties under visible light by loading rGO/ZnO. The changes made in the revised version have improved the quality of the manuscript and may be accepted for publications after the minor corrections, as suggested below.

1. Figure 3-Photocatalytic degradation of RhB by air filter: The Labels on Figures mention Black Control. Is it Blank/control? Please correct it.
2. Fig. 8b: UV-DRS analysis for the rGZ2 and GZ air filters. Please clarify if the Y-axis shows absorbance or FR Function. It cannot be intensity.
3. Fig. 8: With the introduction of rGO, the band gap energy is reduced from 3.0. to 2.64 eV due to the cooperative interface interaction between ZnO and rGO.

This is not true. There is no way rGO can change the band gap of ZnO. The author are suggested to remove such a claim and only put that the rGO/ZnO shows extended absorption as compared to only ZnO

4. The experimental procedure for the synthesis of ZnO is sol-gel processing followed by calcination (400 °C) and cannot be called "in-situ pyrolysis growth at low temperature" as Pyrolysis involves the thermal decomposition of materials at elevated temperatures in an inert atmosphere and 400 °C is not that low.
Please remove the term pyrolysis and write it as "sol-gel processing followed by calcination.

For example, in the abstract, "Herein, a versatile air filter was prepared with loading the reduced graphener and Zinc oxide on the superfine glass fiber with the three-dimensional network structure by in-situ sol-gel process followed by calcination."

5. Figure S1 is not mentioned in the main text.

Review form: Reviewer 3

Is the manuscript scientifically sound in its present form?

Yes

Are the interpretations and conclusions justified by the results?

Yes

Is the language acceptable?

Yes

Do you have any ethical concerns with this paper?

No

Have you any concerns about statistical analyses in this paper?

No

Recommendation?

Accept as is

Comments to the Author(s)

The paper is well revised and can be published as is.

Decision letter (RSOS-202285.R1)

Dear Dr Zhai

On behalf of the Editors, we are pleased to inform you that your Manuscript RSOS-202285.R1 "A superfine glass fiber air filter with rapid response to photocatalytic antibacterial properties undervisible light by loading rGO/ZnO" has been accepted for publication in Royal Society Open Science subject to minor revision in accordance with the referees' reports. Please find the referees' comments along with any feedback from the Editors below my signature.

Please submit your revised manuscript and required files (see below) no later than 7 days from today's (ie 19-Jul-2021) date. Note: the ScholarOne system will 'lock' if submission of the revision is attempted 7 or more days after the deadline. If you do not think you will be able to meet this deadline please contact the editorial office immediately.

on behalf of Professor R. Kerry Rowe (Subject Editor)
openscience@royalsociety.org

Associate Editor Comments to Author:

Thank you for thoroughly revising your paper. A few remaining modifications are needed, but once those are incorporated, the paper may be acceptable for publication.

Reviewer comments to Author:

Reviewer: 1

Comments to the Author(s)

The authors have modified the manuscript according to the reviewers' comments, which can be accepted directly.

Reviewer: 2

Comments to the Author(s)

A superfine glass fiber air filter with rapid response to photocatalytic antibacterial properties under visible light by loading rGO/ZnO

Manuscript No: RSOS-202285-R1

I reviewed the revised manuscript article entitled, "A superfine glass fiber air filter with rapid response to Photocatalytic antibacterial properties under visible light by loading rGO/ZnO. The changes made in the revised version have improved the quality of the manuscript and may be accepted for publications after the minor corrections, as suggested below.

1. Figure 3-Photocatalytic degradation of RhB by air filter: The Labels on Figures mention Black Control. Is it Blank/control? Please correct it.
2. Fig. 8b: UV-DRS analysis for the rGZ2 and GZ air filters. Please clarify if the Y-axis shows absorbance or FR Function. It cannot be intensity.
3. Fig. 8: With the introduction of rGO, the band gap energy is reduced from 3.0. to 2.64 eV due to the cooperative interface interaction between ZnO and rGO.

This is not true. There is no way rGO can change the band gap of ZnO. The author are suggested to remove such a claim and only put that the rGO/ZnO shows extended absorption as compared to only ZnO

4. The experimental procedure for the synthesis of ZnO is sol-gel processing followed by calcination (400 °C) and cannot be called "in-situ pyrolysis growth at low temperature" as Pyrolysis involves the thermal decomposition of materials at elevated temperatures in an inert atmosphere and 400 °C is not that low.

Please remove the term pyrolysis and write it as "sol-gel processing followed by calcination.

For example, in the abstract, "Herein, a versatile air filter was prepared with loading the reduced graphener and Zinc oxide on the superfine glass fiber with the three-dimensional network structure by in-situ sol-gel process followed by calcination."

5. Figure S1 is not mentioned in the main text.

Reviewer: 3
Comments to the Author(s)

The paper is well revised and can be published as is.

===PREPARING YOUR MANUSCRIPT===

===PREPARING YOUR REVISION IN SCHOLARONE===

<https://royalsociety.org/journals/authors/author-guidelines/#supplementary-material> to include a suitable title and informative caption. An example of appropriate titling and captioning may be found at https://figshare.com/articles/Table_S2_from_Is_there_a_trade-off_between_peak_performance_and_performance_breadth_across_temperatures_for_aerobic_scops_in_teleost_fishes_/3843624.

Author's Response to Decision Letter for (RSOS-202285.R1)

See Appendix B.

Decision letter (RSOS-202285.R2)

Dear Dr Zhai,

I am pleased to inform you that your manuscript entitled "A superfine glass fiber air filter with rapid response to photocatalytic antibacterial properties undervisible light by loading rGO/ZnO" is now accepted for publication in Royal Society Open Science.

on behalf of Prof R. Kerry Rowe (Subject Editor)
openscience@royalsociety.org

Appendix A

Dear Editor and Reviewers:

Thank you very much for your letter and for the reviewers' comments concerning our manuscript entitled "A superfine glass fiber air filter with rapid response to photocatalytic antibacterial properties under visible light by loading rGO/ZnO" (ID: RSOS-202285).

These comments are very helpful for revising and improving our manuscript. We have studied these comments carefully and made corrections which we hope meet with approval. Revised contents are marked in red in the revised manuscript. The main corrections in the manuscript and the responses to the reviewers' comments are as following:

Responses to the reviewers' comments:

Reviewer #1:

1. Question: The detailed calculation of surface area of the samples-GF, rGZ1 and rGZ2 measured by the BET method should be explained.

Response: Thank you very much for your good advice. Brunauer-Emmett-Teller (BET) surface areas were determined by a multipoint BET method using the adsorption data in a relative pressure (P/P_0) range of 0.05-0.35 obtained by Gemini VII 5.03 (Micromeritics Instrument Corp) (marked in red). The detailed calculation of the BET of the sample is based on the following equation:

$$\frac{P}{V/(P_0 - P)} = \frac{1}{V_m \cdot C} + \frac{C - 1}{V_m \cdot C} \cdot (P/P_0)$$

$$S_g = \frac{V_m \cdot N \cdot A_m}{22400 \cdot W} \times 10^{-18}$$

P: Adsorbent partial pressure; P_0 : Saturated vapor pressure of adsorbent; V_m : saturated adsorption capacity of nitrogen molecule monolayer under standard conditions; C: adsorption capacity coefficient; W: the mass of the sample, N: 6.02×10^{23} ; A_m : the maximum equivalent cross-sectional area of nitrogen molecules.

The test results are shown below, with the introduction of zinc oxide and rGO, the

BET area of the air filter has increased from 2.3641 to 9.7256 m²/g. The increase of the specific surface area can effectively increase the photocatalytic activation performance of the sample to a certain extent.

2. Question: The adhesion of rGZ and ZnO on glass fiber need be provided.

Response: Thank you very much for the good advice. We tried to verify the binding force of zinc oxide and rGO on the surface of glass fiber from the change of the surface morphology of the sample (marked in red). The result is shown in Figure 12b. After high-speed centrifugation, the rGZ2 air filter still maintains a tight covering, which reflects the structural stability of our synthetic rGZ2 air filter.

Figure 12. The fractional efficiency filter test of the s-GF and rGZ2 air filter (a); the morphology of the sample under the scanning electron microscope before and after centrifugation(b).

Reviewer #2:

1. Question: No air-filtration experiments were performed as per the title of the manuscript.

Response: We have made correction according to the Reviewer's comments in Figure 12a (marked in red). The new discussion is as following. For air-filtration experiments,

it can be seen from the Figure 12a that whether the surface of s-GF is loaded with ZnO/rGO-based particles, the filtration efficiency remains at 99.9% under the test of 5 kinds of particles with different diameters. The MPPS of the s-GF and the rGZ2 air filter is 0.12216 and 0.12104 μm , respectively. This result indicates that the loading of composite photocatalytic particles on the surface of glass microfiber does not reduce the filtration performance of the air filter. The air-filtration experiments direct that the rGZ2 air filters still maintain a stable and effective interception for aerosol particles.

Figure 12. The fractional efficiency filter test of the s-GF and rGZ2 air filter (a); the morphology of the sample under the scanning electron microscope before and after centrifugation(b).

2. Question: Figure 8: An arbitrary mechanism is proposed without providing experimental evidences of the data included in Fig. 8.

Response: I am sorry for the lack of experiments on the photocatalytic mechanism. The discussion of the photocatalytic mechanism is shown (marked in red). As discussed in Question 1, the air filtration performance has been verified by testing. In order to further reveal the active groups during the photocatalytic degradation and antibacterial process of rGZ2, the ESR spectra of the free radicals captured by DMPO in rGZ2 was studied. As shown in Figure 9, after 4 minutes, 8 minutes and 12 minutes of irradiation, the six characteristic peaks of DMPO- $\text{O}_2^{\cdot-}$ (Figure 9b) and four characteristic peaks of DMPO- $\cdot\text{OH}$ (Figure 9c) can be clearly seen in rGZ2 air filter. In contrast, rGZ2 in the dark did not find a significant signal peak of free radicals, indicating that the air filter did not

generate free radicals in the absence of light. This reveals that the rGZ2 air filter produces two free radicals DMPO-•OH and DMPO -O₂^{•-} under light conditions, and this also plays an important role in the research on the mechanism of photocatalytic degradation of organic dyes and rapid sterilization.

Figure 9. ESR spectra of (a) O₂^{•-} and (b) •OH obtained from the rGZ2 air filter after different exposure times under simulated light.

3. Question: The reasons for difference in photocatalytic/antibacterial activity are not discussed with adequate justification and assigning the higher activity of rGZ2 to surface area effect (and why it has high surface area?) is not sufficient. Other claim that rGO reduced the e⁻-h⁺ recombination and increase absorption in visible region need experimental evidence (PL measurements, and diffuse reflectance measurement (band gap evaluation). That rGO enhances dyes adsorption needs to be justified with experimental evidence. That rGO leads to a decrease in ZnO particle size may be explained better providing PSD histograms in ESI.

Response: Thank you very much for the good advice. Based on the reasons for the difference in the photocatalytic activity of the samples, as the reviewer said, we have conducted a supplementary experiment: PL and UVDRS tests on GZ and rGZ2 air filter in Figure 8(marked in red). The test results are discussed as follows. Figure 8a shows the photoluminescence (PL) emission (excited at 325 nm) spectra of the GZ and rGZ2 air filters. A strong ultraviolet emission peak at 386 nm corresponding to near band-edge emission is attributed to the excitonic recombination of a hole in the valence band and an electron in the conduction band. The intensity of the PL emission band can be

used to reflect the lifetime of electron-hole pairs because PL comes from the recombination of electron-hole pairs after the photocatalyst is irradiated with light. Electrons excited by light on zinc oxide can be transported to the interface of rGO, which effectively reduces the recombination of photogenerated electron-hole pairs and thus weakens the emission peak of the rGZ2. The weaker emission peak indicates a greater separation rate of photogenerated electron-hole pairs in the material, which further reflects the stronger photocatalytic activity of the rGZ2. Figure 8b depicts the UV absorption spectra of GZ and rGZ air filters recorded in the range of 200-800 nm. It can be seen that the GZ only targets the absorption of ultraviolet light with a wavelength below 380 nm. Different from GZ, the light absorption range of rGZ2 air filter paper is obviously red-shifted, showing strong absorption ability in the visible light region. The band gap energies of GZ and rGZ2 calculated from the Tauc's equation: $(\alpha h\nu)^2 = A (h\nu - E_g)^2$ are shown in Figure 8c. With the introduction of rGO, the band gap energy is reduced from 3.0. to 2.64 eV due to the cooperative interface interaction between ZnO and rGO. This means that all photon energy of visible light can be absorbed by GZ, so the introduction of rGO in GZ can greatly expand the light absorption range and improve the photocatalytic activity of the air filter. The spectroscopic results of PL and UV-DRS indicate that the photocatalytic activity of rZG2 should indeed be higher than that of the GZ air filter.

Figure 8. The room temperature PL emission spectra of the GZ and rGZ2 is shown in (a). UV-DRS analysis for the rGZ2 and GZ air filters (b). A plot of $(\alpha h\nu)^2$ versus the band gap energy (eV) for GZ and rGZ2 filters (c).

For the second question, thank you very much for your careful work. For rGO to enhance solution absorption, as shown in Figure 2, we think it can be explained from the photocatalytic degradation experiment of the RhB solution. The discussion is as following. By comparing the dark adsorption results of s-GF (Figure 2b), GZ (Figure 2c), rGZ1(Figure 2d) samples within 30 minutes. We can also calculate the adsorption rates of GZ, rGZ1, and rGZ2 samples to be 2%, 3% and 15%, respectively. From this result, it can be seen that the addition of rGO improves the adsorption capacity of the sample to the solution because the content of ZnO is consistent in the air filters.

For the third question, considering the Reviewer's suggestion, we consider that the sample is a three-layer mesh composite structure based on rGO/ZnO/S-GF. We are sorry that we cannot provide this PSD histogram. But we can still try to analyze the reduction of ZnO particle size by rGO from the results of scanning electron microscopy. The morphology of nanoparticles on the surface of GZ air filters was enlarged as shown in Figure 4e, it found that the average diameter nanoparticles was between 50 and 100 nm

and distributed relatively uniform. Compared with Figure 4e, Figure 4f shows the average size of ZnO reaching to the range of 20-50 nm when the rGO seed layer is loaded on the surface of s-GF, indicating that the addition of rGO can solve the agglomeration problem and also refine the crystal grains of ZnO.

4. Question: The experimental procedure for the synthesis of ZnO is sol-gel processing followed by calcination (400 °C) and cannot be called “in-situ pyrolysis growth at low temperature” as Pyrolysis involves the thermal decomposition of materials at elevated temperatures in an inert atmosphere and 400 °C is not that low. Moreover, the Material and Method section reports the use of Zinc acetate dehydrate (material) and then anhydrous Zn-acetate (experimental).

Response: Thank you very much for your reminding. We have re-written this part according to the Reviewer’s suggestion in chapter 3.2 (marked in red). The experimental procedure for the synthesis of ZnO is called “in-situ pyrolysis growth based on sol-gel process”. We have also made revised to the uniformity of the experimental materials.

5. Question: Relevant experimental details (such as charge correction in XPS etc) are missing.

Response: Thank you very much for your guidance. The correction of XPS is performed on the 284.8ev potential of carbon, we have revised some relevant experimental details and supplemented the experimental methods under the reviewers’ suggestions in chapter 3.3 (marked in red).

6. Question: The antibacterial tests are reported to be performed under LED illumination (see the conclusion part) but there is no mention of it in the experimental part which actually reports the use of 500 W Xenon as light source in the photocatalytic tests.

Response: Thank you very much for your careful work. For the LED lamp used in the antibacterial experiment, we have made a supplement in chapter 3.5 (marked in red).

The 60 W LED light is used as the light source and the plate colony counting method were used in this test.

7. Question: The theoretical fit on experimental XPS data (or fitting residue) is missing, making the fitting less reliable. The assignment of 283.36 eV photoemission peak to C-O-Zn needs reevaluation as the cited references (30, 31) say nothing about it.

Response: We are very sorry for our negligence of experimental XPS data. As the reviewer mentioned in Question 5, our charge correction for XPS is not accurate enough. We have re-written this part according to the Reviewer's suggestion in section 4.4 (marked in red). The high-resolution scan of C 1s spectra (Figure 7c) shows three different peaks at 284.8, 286.2 and 290.2 eV, which are designated as C-C, C-O and C=O bonds of the air filter, respectively.

Figure 7. X-ray photoelectron spectroscopy spectra of rGZ2 (a) Survey (b) Zn 2p (c) C 1s (d) O 1s

8. Question: The claim that rGO/ZnO loaded on the surface of s-GF is not toxic to bacterial cells is not totally true. ZnO may show some antibacterial properties in dark (doi: 10.1016/j.jiec.2016.10.013) but no such effects are seen in Fig. 9.

Response: Thank you very much for providing the excellent work. We have added the articles in the revised manuscript (Reference 48) (marked in red). The antibacterial activity of ZnO has indeed been confirmed in Figure 11g-h. The result is shown below. In the article, the results of this study provide empirical evidence that ZnO nanoparticles at low concentrations could pose significant risks to environmental microorganisms even under dark conditions. The average colony numbers of GZ samples against *E. coli* and *Staphylococcus aureus* under dark conditions were 652 and 657 CFU/mL, respectively. This also allows us to have a better understanding of the actual application about the air filter in the future.

Figure 11. Images of antibacterial activities of (g-h) the GZ and (i-j) rGO/s-GF air filter.

9. Question: Finally, the English of the manuscript require proof reading as there are dozens of words with no spacing in between and other grammatical error.

Response: Thank you very much for your reminding. We have made correction according to the Reviewer's comments such as the section 3.2, 3.4 and 3.5(marked in red).

Reviewer #3:

1. Question: In summary section, the author stated the importance and challenges of high-performance air filter, and this article also focuses on superfine glass fiber air filters, but there is no air purification performance data in the article.

Response: Thank you very much for your guidance. We have made correction according to the Reviewer's comments in Figure 12a (marked in red). The new discussion is as following.

For air-filtration experiments, it can be seen from the Figure 12a that whether the surface of s-GF is loaded with ZnO/rGO-based particles, the filtration efficiency remains at 99.9% under the test of 5 kinds of particles with different diameters. The MPPS of the s-GF and the rGZ2 air filter is 0.12216 and 0.12104 μm , respectively. This result indicates that the loading of composite photocatalytic particles on the surface of glass microfiber does not reduce the filtration performance of the air filter. The air-filtration experiments direct that the rGZ2 air filters still maintain a stable and effective interception for aerosol particles.

Figure 12. The fractional efficiency filter test of the s-GF and rGZ2 air filter (a); the morphology of the sample under the scanning electron microscope before and after centrifugation(b).

2. Question: Why does rGZ2 air filter have the best photocatalytic RhB decolorization performance? The content of ZnO and rGO is not the highest, and the author should give a clear explanation.

Response: Thank you very much for your guidance. We have measured the PL and DRS spectra of GZ and rGZ in Figure 8 (marked in red). The intensity of the PL emission band can be used to reflect the lifetime of electron-hole pairs because PL comes from the recombination of electron-hole pairs after the photocatalyst is irradiated with light. Electrons excited by light on zinc oxide can be transported to the interface of rGO, which effectively reduces the recombination of photogenerated electron-hole pairs and thus weakens the emission peak of the rGZ2 in Figure 8a (marked in red). At the same time, we have supplemented the PL emission spectra of rGZ1, rGZ2, rGZ3, rGZ4 air filters in Figure S1. The weaker emission peak indicates a greater separation rate of photogenerated electron-hole pairs in the material, which further reflects the stronger photocatalytic activity of the rGZ2.

Figure 8. The room temperature PL emission spectra of the GZ and rGZ2 is shown in (a). UV-DRS analysis for the rGZ2 and GZ air filters (b). A plot of $(\alpha h\nu)^2$ versus the band gap energy (eV) for GZ and rGZ2 filters (c).

Figure S1. The room temperature PL emission spectra of the rGZ1, rGZ2, rGZ3 and rGZ4

3. **Question:** What was the author use in section 3.1 and 3.2, anhydrous zinc acetate or zinc acetate dihydrate?

Response: We are very sorry for the negligence expressed in the material. We have made revised to the uniformity of the experimental materials in section 3.1 and 3.2 (marked in red).

4. **Question:** In section 4.2, “As can be seen in Figure 4a, the surface of the s-GF without any loading treatment is very smooth and presents a three-dimensional network structure, which can well explain the cause of excellent air filtration performance of the original s-GF.” This part lack of data support.

Response: We are sorry for the lack of data support. The air filtration test of the s-GF sample is shown in Figure 12a (marked in red). The s-GF shows 99.8% air filtration efficiency and the MPPS of the s-GF is $0.12216 \mu\text{m}$, which can well explain the cause of excellent air filtration performance of the original s-GF.

Figure 12. The fractional efficiency filter test of the s-GF and rGZ2 air filter (a); the morphology of the sample under the scanning electron microscope before and after centrifugation(b).

5. Question: In introduction section, “Furthermore, adding ZnO nanoparticles can also inhibit the agglomeration between rGO layered structures to a certain extent.” However, it is pointed out that the addition of rGO can solve the agglomeration of ZnO. (in section 4.2). It is contradictory.

Response: Thank you very much for your guidance. We are sorry for this unclear explanation about the dispersion of ZnO and rGO. We have made correction according to the Reviewer’s comments (marked in red) (in section 4.2). In introduction section, through some excellent work, we introduced the improvement of the photocatalytic performance of ZnO by the introduction of rGO. However, agglomeration may occur due to the strong electrostatic force between the rGO sheets. In the experimental part (in section 3.2), we also introduced that by repeatedly immersing s-GF in rGO dispersion and ZnO precursor solution to increase the loading capacity, it also improved the dispersibility of rGO due to the formed rGO/ZnO structure. In the discussion part, we also expressed that rGO can improve the agglomeration problem of ZnO particles. These results are also confirmed by subsequent characterization methods such as scanning electron microscopy in Figure 2. It is concluded that the dispersibility of rGO

and ZnO is improved through the mutual synergy between the two.

6. Question: It is reported that GO also has antibacterial performance. The author should give the antibacterial performance of the rGO/s-GF air filter to better explain the sterilization mechanism of the filters.

Response: Thank you very much for your guidance. The antibacterial activity of ZnO has indeed been confirmed in Figure 11i-j. The result is shown below. The average colony numbers of rG samples against *E. coli* and *S. aureus* under dark conditions were 652 and 725 CFU/mL, respectively. The rG filter was cultured for 4 h under simulated visible light irradiation and its average colony count against *E. coli* and *S. aureus* was 57 and 31 CFU/mL, respectively. This reveals that rGO has the ability to inactivate bacteria, but it still grows bacterial colonies on the glass fiber. This also has certain risks for the practical application of air filters, so it is necessary to combine ZnO/rGO to produce synergistic effects. This also allows us to have a better understanding of the actual application about the air filter in the future.

Figure 11. And images of antibacterial activities of (g-h) the GZ and (i-j) rGO/s-GF air filter.

7. Question: Typos and format errors in the manuscript should be corrected. n, e.g., in section 3.2, “... distilled water and then 30mL...” , “... with a heating rate of 5°C /min” , missing spaces.

Response: Thank you very much for your reminding. We have made correction according to the Reviewer's comments such as the section 3.2, 3.4 and 3.5 (marked in red).

Appendix B

Dear Editor and Reviewer#2:

We appreciate the thorough evaluation of our study entitled “A superfine glass fiber air filter with rapid response to photocatalytic antibacterial properties under visible light by loading rGO/ZnO” (ID: RSOS-202285) by the reviewers and editor.

These comments are very helpful for revising and improving our manuscript. We have studied these comments carefully and made corrections which we hope meet with approval. The added contents are marked in red in the revised manuscript. All corrections responding to reviewer#2’s comments in current manuscript are as following:

Reviewer #2:

Question 1: Figure 3-Photocatalytic degradation of RhB by air filter: The Labels on Figures mention Black Control. Is it Blank/control? Please correct it.

Response: Thank you very much for your careful work. The condition in Figure 3 should be dark adsorption. We have completed the correction for the label of Figure 3.

Figure 3-Photocatalytic degradation of RhB by air filter

Question 2: Fig. 8b: UV-DRS analysis for the rGZ2 and GZ air filters. Please clarify if the Y-axis shows absorbance or FR Function. It cannot be intensity.

Response: Thank you very much for your reminding. The Y-axis should show absorbance in Fig. 8b. We have completed the targeted modification and the result is shown in the figure below.

Figure 8. The room temperature PL emission spectra of the GZ and rGZ2 is shown in (a). UV-DRS analysis for the rGZ2 and GZ air filters (b). A plot of $(ahv)^2$ versus the band gap energy (eV) for GZ and rGZ2 filters (c).

Question 3: Fig. 8: With the introduction of rGO, the band gap energy is reduced from 3.0. to 2.64 eV due to the cooperative interface interaction between ZnO and rGO. This is not true. There is no way rGO can change the band gap of ZnO. The author are suggested to remove such a claim and only put that the rGO/ZnO shows extended absorption as compared to only ZnO.

Response: Thank you very much for the good advice. It is really true as Reviewer suggested that rGO does not change the band gap of ZnO. We have re-written this part according to your suggestion marked in red in chapter 4.5. The new discussion is as follows: It can be seen the rGO/ZnO shows extended absorption as compared to only ZnO due to the cooperative interface interaction between ZnO and rGO.

Question 4: The experimental procedure for the synthesis of ZnO is sol-gel processing followed by calcination (400 °C) and cannot be called “in-situ pyrolysis growth at low temperature” as Pyrolysis involves the thermal decomposition of materials at elevated temperatures in an inert atmosphere and 400 °C is not that low. Please remove the term pyrolysis and write it as “sol-gel processing followed by calcination.

Response: Thank you very much for your reminding. We have re-written this part according to your suggestion marked in red in chapter 3.2. The experimental procedure

for the synthesis of ZnO is called “in-situ sol-gel process followed by calcination”.

Question 5: Figure S1 is not mentioned in the main text.

Response: We have revised some relevant details under your suggestion marked in red in chapter 4.1. The specific discussion is as follows: the PL emission spectra of rGZ1, rGZ2, rGZ3, rGZ4 air filters was shown in Figure S1. The weaker emission peak indicates a greater separation rate of photogenerated electron-hole pairs in the material, which further reflects the stronger photocatalytic activity of the rGZ2.

Figure S1. The room temperature PL emission spectra of the rGZ1, rGZ2, rGZ3 and rGZ4